# Observation of Collisional De-Excitation Phenomena in Plutonium

**Andrea Raggio \***  , **Ilkka Pohjalainen**  and **Iain D. Moore**

Accelerator Laboratory, Department of Physics, University of Jyväskylä, FI-40014 Jyväskylä, Finland;
ilkka.pohjalainen@jyu.fi (I.P.); iain.d.moore@jyu.fi (I.D.M.)
**\*** Correspondence: andrea.a.raggio@jyu.fi; Tel.: +358-04-57-833-2669

**Abstract:** A program of research towards the high-resolution optical spectroscopy of actinide elements for the study of fundamental nuclear structure is currently ongoing at the IGISOL facility of the University of Jyväskylä. One aspect of this work is the development of a gas-cell-based actinide laser ion source using filament-based dispensers of long-lived actinide isotopes. We have observed prominent phenomena in the resonant laser ionization process specific to the gaseous environment of the gas cell. The development and investigation of a laser ionization scheme for plutonium atoms is reported, focusing on the effects arising from the collision-induced phenomena of plutonium atoms in helium gas. The gas-cell environment was observed to greatly reduce the sensitivity of an efficient plutonium ionization scheme developed in vacuum. This indicates competition between resonant laser excitation and collisional de-excitation by the gas atoms, which is likely being enhanced by the very high atomic level density within actinide elements.

**Keywords:** collisional de-excitation; actinide elements; resonance laser ionization; gas cell





## 1. Introduction

In radioactive ion beam (RIB) facilities based on the isotope separator on-line (ISOL) technique, laser resonance ionization is a widely used method for producing isobarically pure ion beams in combination with mass separation, thanks to its high efficiency and selectivity. Its implementation in facilities that use gas-cell-based techniques for ion thermalization and extraction, such as the ion guide isotope separator on-line (IGISOL) [1] technique at the University of Jyväskylä, the former Leuven isotope separator on-line (LISOL) in Louvain-La-Neuve [2], or the KEK isotope separation system at RIKEN [3], have led to the development of the in-gas-cell laser resonance ionization technique. The combination of this method with the fast gas-cell extraction time and the chemical non-selectivity of the used noble gasses allows for the production and study of refractory elements, which are notoriously challenging for the traditional ISOL technique. Laser resonance ionization is based on a multi-step excitation scheme, utilizing the unique atomic-level fingerprint of the desired element. Resonance ionization spectroscopy is a variant of the laser resonance ionization technique, whereby the mass-separated ion yield is measured as a function of the wavelength of one of the lasers used in the excitation process. By measuring the isotope shift and splitting (hyperfine structure) of the optical resonance, a comprehensive probe is provided into underlying nuclear properties, including nuclear spins, sizes and shapes [4].

When laser resonance excitation and ionization is applied in a gas-filled environment, a series of mechanisms start to play a detrimental role. For example, the spectral resolution of atomic resonance in a gas cell suffers from pressure broadening and shift due to collisions with the buffer gas atoms. Typical operating pressures of up to 500 mbar limit the resolution to a few GHz, sufficient to mask the hyperfine structure of lighter elements. Nevertheless, in-gas cell spectroscopy has the advantage of being a very sensitive method and optical spectroscopy can be performed on isotopes produced with small cross-sections. A recent highlight has been the successful demonstration of on-line, single-atom-at-a-time, resonance

ionization spectroscopy of nobelium in an argon-filled gas cell [5]. A second, less-well-documented effect of the gas-filled environment, also arising from collisional interactions, is the potential reduction in the laser ionization efficiency through collisional de-excitation. This effect, which is of topical interest and the focus of this work, may be enhanced in elements with high atomic-level densities such as the lanthanides or actinides.

*Collision-Induced Population Transfer*

Collisions between atoms can cause changes in the electron population between atomic levels. A well known example of this is the Boltzmann-distributed population of low-lying levels in atoms emerging from thermal collisions with buffer gas atoms. Similarly, collisions with an excited atom can cause the excitation to be transferred to a different level or to the collision partner. In this work, we are interested in the collisional de-excitation of the resonantly excited states in plutonium and the competition with laser-induced resonant transitions. The dynamics of collisional de-excitation are examined below to give a qualitative understanding of the collision process in plutonium. However, as the atomic structure of heavy elements can be exceedingly complicated, the mechanism of collisional de-excitation is introduced here with the help of simpler systems.

Traditionally, collisional phenomena have been studied mostly in alkali and alkaline-earth atoms and ions with noble gasses [6–11], since only few loosely bound outer electrons participate in the collision, resulting in an easier theoretical modeling. Other elements studied in this context are reported, for example, in [12–16]. In general, it has been observed that the rate of population transfer is higher for heavier noble gasses.

A classic example and practical application of collisional excitation transfer is the helium–neon (HeNe) laser system. The helium atoms are excited from the ground state to higher-lying excited states by inelastic collisions with energetic electrons, among them the first two metastable states $1s2s\ ^3S_1$ and $1s2s\ ^1S_0$. Due to a fortuitous near degeneracy between the helium metastable states and excited states in neon atoms, collisions result in an efficient and selective transfer of excitation energy from the helium to neon, as reported, for example, in [17]. The subsequent decays of the excited states in neon are responsible for the characteristic emission wavelengths of the laser system.

Instead of transferring the energy of the colliding system to the collision partners, a second scenario is to convert the excitation energy of the colliding atoms into translational energy. This can be understood using the barium–argon system as an example, as barium has a rather simple electronic structure compared with heavier elements [18–21]. The collisional de-excitation process can be understood by looking at the potential curves of the diatomic system, illustrated in Figure 1, formed by barium configured in different excitation states and argon in the $^1S_0$ ground state. If such curves have crossing points at given interatomic distances, marked with a circle in Figure 1, and if the symmetry of the crossing states allows the formation of an avoided crossing, then an inelastic collision can occur, transferring the population to the lower excited state and converting the energy difference into the translational energy of the system. For this specific example, barium atoms, excited to the $6s6p\ ^1P_1^o$ state, for example, via laser excitation, are collisionally de-excited to the $6s6p\ ^3P_2^o$ level through inelastic collisions with argon atoms. The crossing happens between the attractive $^1\Pi_1$ potential curve and the repulsive $^3\Sigma_1$ potential curve, the latter adiabatically correlated with the $^3P_2^o$ barium level.

The rate of collisional de-excitation can be comparable to the optical de-excitation that occurs via spontaneous emission. For example, the rate of collisional de-excitation in barium of the $6s8p\ ^1P_1^o$ level by helium collisions at 833 K to the $6s7d\ ^3D_j$ multiplet has been measured to be $\sim 3 \times 10^{-9}\ \mathrm{cm^3/s}$ [18], while the rate of de-excitation from the $3s^2 3p^4\ ^1D_2$ state in atomic sulphur in argon at 300 K has been measured to be $1.4 \times 10^{-11}\ \mathrm{cm^3/s}$ [22]. With a typical gas cell pressure (either helium or argon) of 100 mbar, these reaction rates result in depopulation rates of about $3.3 \times 10^9\ \mathrm{1/s}$ and $4 \times 10^7\ \mathrm{1/s}$ for barium and sulphur, respectively, comparable to a large $A_{if}$ Einstein coefficient.

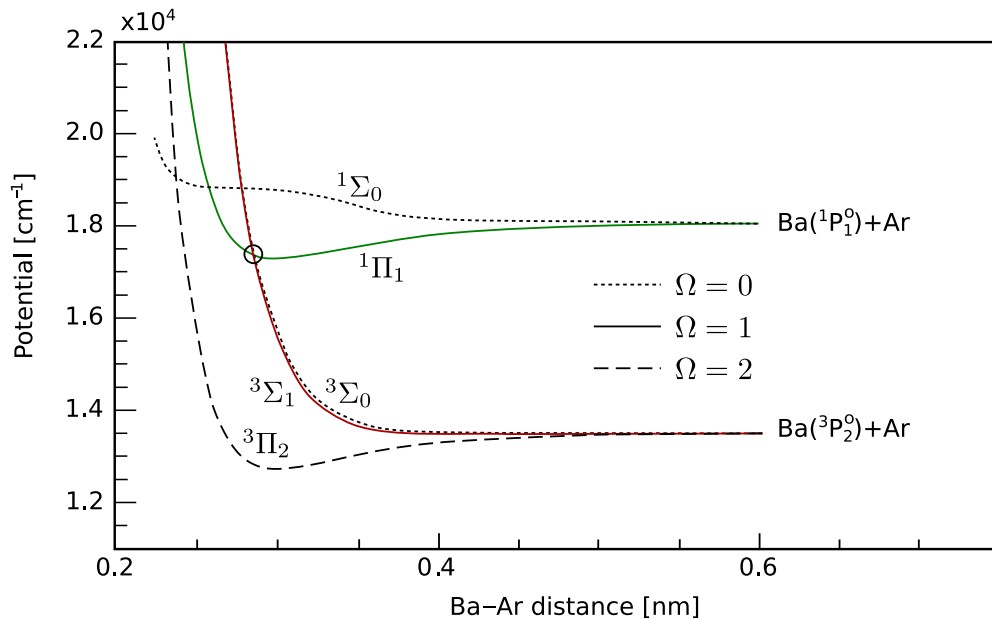

**Figure 1.** Diabatic potential curves in the barium–argon system. The level crossing indicated by a circle is responsible for the population transfer between the $^1P_1^o$ and $^3P_2^o$ levels of barium. Adapted with permission from [21]. 1992, American Physical Society.

In heavier elements, namely the actinide region, a non-negligible collisional de-excitation cross section is expected due to the high density of states compared with the simpler alkali or alkaline-like elements. Collisional phenomena have been observed in thorium [23] and exploited in laser spectroscopy measurements of singly ionized $Th^+$ ions [24,25]. The laser excitation of thorium from the ground state to a specific excited state can be inefficient as spontaneous decays increase the population of metastable (dark) states, which are not probed by the laser light. The use of buffer gasses such as hydrogen and helium in radiofrequency traps helps to redistribute the population of these states to the ground state via inelastic quenching collisions, improving the laser ionization rate.

Collisional phenomena were observed and also studied in nobelium and its chemical homologue ytterbium [26,27]. A two-step resonance ionization process was used in a buffer gas cell: the first excitation step to resonantly excite nobelium from the $^1S_0$ atomic ground state to the $^1P_1$ level, while the second step drove a transition to a Rydberg state, which was subsequently ionized either by residual laser light, black-body radiation or collisional processes. Importantly, analysis of the Rydberg states revealed contributions from two intermediate states, the singlet $^1P_1$ level and a longer-lived state lying ∼300 cm$^{-1}$ below the $^1P_1$ level. This lower-lying state, assigned as the $^3D_3$ level, was populated via a fast quenching from the $^1P_1$ level due to collisions with buffer gas atoms.

Here, we report evidence of such collision-induced phenomena for the case of plutonium via laser resonance ionization studies. Although collisional effects of actinide elements with noble gasses have been reported, especially for low-lying excited states, collisional de-excitation effects remain mostly unknown in the excited states.

## 2. Experimental Method

This work was performed in the context of the development of a new gas cell for the laser resonance ionization of long-lived actinide isotopes that can be produced in sufficient quantities at research reactors and transported to facilities elsewhere [28]. In collaboration with the Nuclear Chemistry Department of the University of Mainz, samples of $^{238-240,242}$Pu and $^{244}$Pu isotopes were electrolytically deposited onto a tantalum substrate and delivered to Jyväskylä. The filaments, mounted within the gas cell filled with helium at a pressure of 80 mbar, were electrically heated to a temperature between 1000 and 1200 °C. The evaporated plutonium atoms were resonantly ionized with laser light provided by

the FURIOS laser system [29], with broadband titanium–sapphire (Ti:sa) lasers operating at a repetition rate of 10 kHz in both the fundamental emission range as well as frequency doubled. Later, a dedicated grating-based Ti:Sapphire laser was employed in collaboration with Nagoya University, having the unique feature of intracavity second harmonic generation (SHG) [30]. This laser offers a wide-range scanning capability between 380 nm and 440 nm and thus is ideally suited for ionization scheme development.

The different lasers were spatially overlapped, transported to the front-end of the IGISOL target area and focused slowly into the gas cell through a quartz window. A maximum output power of ∼1 W of fundamental infrared (IR) light was available at the entrance to the gas cell. After intra-cavity frequency doubling, ∼660 mW from the standard broadband resonator was available, with ∼105 mW from the grating-based laser. In our studies of a two-step blue–blue ionization scheme, discussed below, both the JYFL Ti:sa laser and the grating-based laser were pumped by separate Nd:YAG lasers to allow for a precise timing control of the laser pulses to a few ns.

The resonant photo-ions were evacuated from the gas cell through an exit hole and guided towards the high-vacuum region of the mass separator using a radiofrequency sextupole ion guide (SPIG) [31]. The ions were accelerated to a potential of 30 kV, mass separated with a nominal mass resolving power of $M/\Delta M = 500$, and detected in the focal plane of the separator using a multichannel plate (MCP) detector. A schematic overview of the experimental setup is shown in Figure 2.

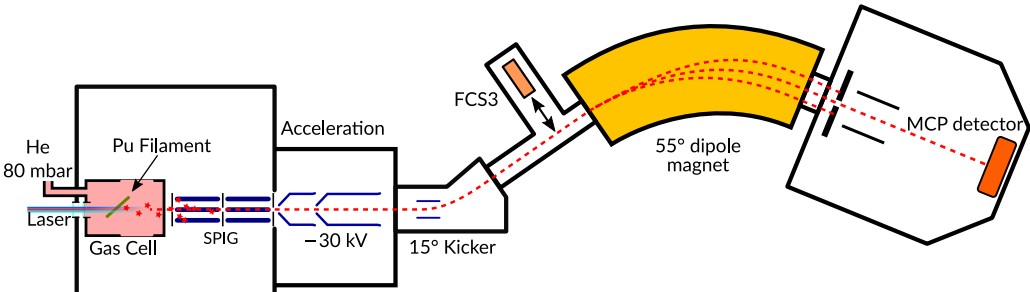

**Figure 2.** Schematic figure of the IGISOL beamline of relevance to this work. The gas cell is located inside the front-end vacuum chamber. Resonant laser ionization of plutonium atoms occurs along the gas cell axis; the ions are guided through a sextupole ion guide and accelerated towards the mass separator. The dipole magnet is set to mass-separate $^{244}$Pu. An MCP detector in the separator focal plane is used to measure the ion counting rate. FC = Faraday cup.

Our earlier work [28] focused on the obtained mass spectra, which provided useful insight into the gas-phase chemistry exhibited by plutonium. The resulting monatomic yields of isotopes were sufficient for high-resolution collinear laser spectroscopy [32], with plutonium currently the heaviest element studied using this technique to date. Since the publication of Ref. [28], further investigations to elucidate a better understanding of the original ionization scheme have been made, using the same methodology as discussed in this section. As the isotopic composition of the samples was not of particular interest, the separator was tuned to the most abundant isotope, $^{244}$Pu. The following section presents the results of the ionization scheme characterization.

## 3. Results and Discussion

Initially, a three-step ionization scheme was tested, using laser radiation at wavelengths of 420.76, 847.26, and 750.24 nm, with the final step resulting in the population of an auto-ionizing (AI) state at 48,898 cm$^{-1}$ (Figure 3). This scheme was originally developed by Raeder and collaborators to selectively ionize plutonium isotopes under vacuum for trace analysis studies of environmental samples [33]. Surprisingly, in the gas-cell environment, the two IR steps did not contribute to the ion count rate. Nevertheless, a frequency scan of the first blue step wavelength presented a clear resonant signal and thus it was hypothesized that excitation and ionization proceeded via a Rydberg state located at around

47,532 cm$^{-1}$, with ionization occurring via atomic collisions with He gas atoms. Only at a substantially reduced first step laser power was a small response to the ion count rate observed, with the addition of the two IR steps.

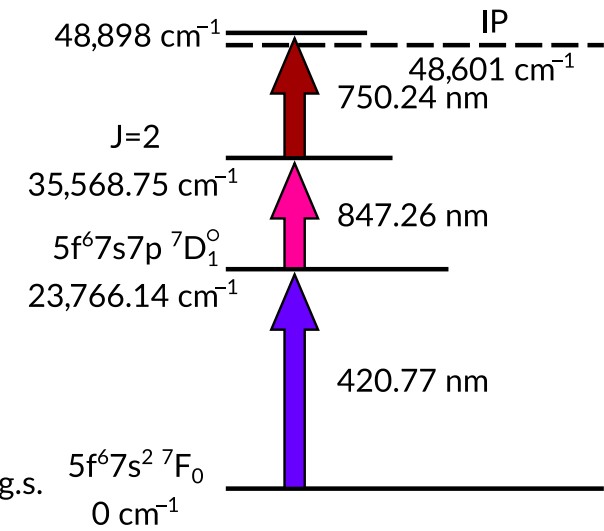

**Figure 3.** Resonance ionization scheme initially used in this work, developed for ultratrace analysis of environmental samples [33]. The three step blue-IR-IR scheme drives the electrons to an auto-ionizing state located at 48,898 cm$^{-1}$.

To explore this hypothesis, a second frequency-doubled Ti:Sapphire laser was implemented instead of the two IR lasers. The wavelength of the second laser was scanned in the vicinity of the first step 420.76 nm transition, with a resonance observed at a wavelength of 422.53 nm which, in combination with the original 420.77 nm, was found to considerably enhance the ionization rate. Interestingly, both transitions were found to ionize plutonium independently, albeit with much reduced count rates. Although there is no known level that can be populated from the ground state by the 422.53 nm laser, the low-lying first excited $^7F_1$ state at 2203.61 cm$^{-1}$ is expected to be thermally populated due to the temperature of the hot filament (and the surrounding helium gas atoms). Excitation would then proceed from this J = 1 level to a known (J = 2) level at 25,870.69 cm$^{-1}$. Both 420.76 and 422.53 nm photons then drive the electron across the ionization potential. The population of the thermally excited state explains the ability of the two lasers to independently ionize plutonium, under the assumption that the ground-state transition is connected to the ionization potential via a high-lying Rydberg state, as previously hypothesized.

We note that the ionization rate with both blue laser transitions was found to be ~5 times greater than the sum of the two ion rates obtained independently. This behavior suggests a connection between the two transitions and we postulate a population of the low-lying state at 2203.61 cm$^{-1}$ from the $^7D_1^o$ level (Figure 3) through collisionally induced de-excitation. If this de-excitation process is fast compared with the original IR transitions or the AI-state lifetime, it could explain the negligible effect of the two IR steps in the original scheme when applied within the buffer gas environment.

To further study this behavior, a third experiment was carried out using the grating-based Ti:Sapphire laser from Nagoya University, combined with a standard broadband frequency-doubled Ti:Sapphire. The latter was tuned to the original ground-state transition at 420.77 nm, while the former was used to perform a wide-range wavelength scan around the region of the previously found 422.53 nm transition. The result of this scan is shown in Figure 4, illustrating the ion count rate in the case of the presence or absence of the 420.77 nm step. One can immediately see the effect of introducing the 420.77 nm laser transition, as the background ion rate is considerably enhanced, which we attribute to the laser constantly ionizing plutonium independent from the grating-based laser, postulated to occur via a potential Rydberg level, as noted earlier.

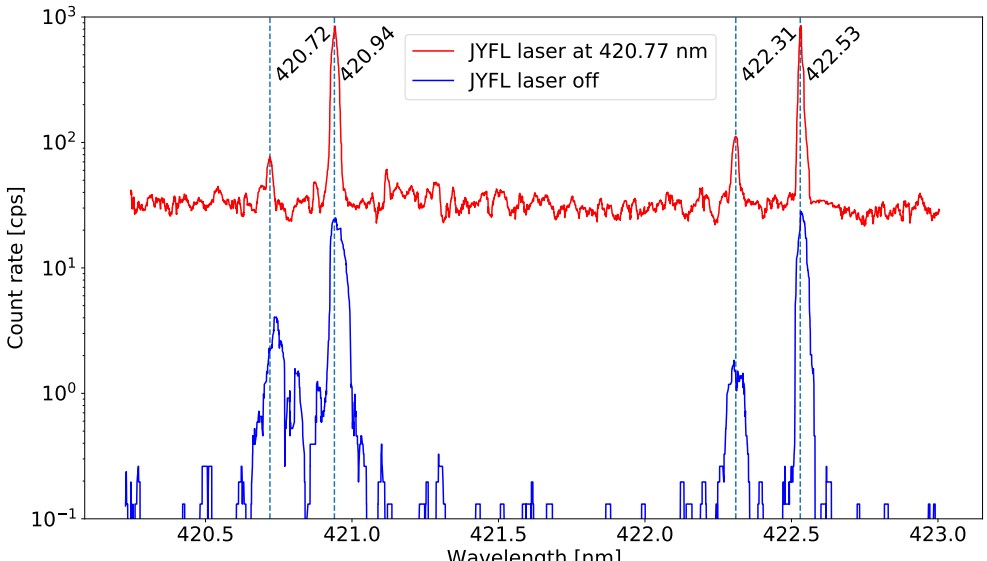

**Figure 4.** Wavelength scan of plutonium obtained from the grating-based Ti:Sapphire with (red) and without (blue) the 420.77 nm laser light produced by the JYFL broadband frequency-doubled Ti:Sapphire laser. The vertical dotted lines indicate the peak maxima according to the upper spectrum to allow for comparisons with the lower spectrum.

In addition to the previously detected transition at 422.53 nm, both spectra indicate the presence of new resonances. The centroid wavelengths were compared with available atomic level data [34], with all possible electric dipole transitions close to the detected resonances considered. A summary of the measured transitions is reported in Table 1 along with the literature assignment, the initial and final energy of the states involved, atomic spins and state configurations. Interestingly, the new transitions all originate from the first few low-lying states in plutonium. As the grating-based laser can excite and ionize the atoms without the presence of the 420.77 nm laser radiation, these low-lying levels must all be thermally populated due to the filament temperature. As an aside, we note that the resolution of the resonances in the lower spectrum of Figure 4 is slightly worse, likely indicating differences in the linewidths of the two lasers.

**Table 1.** List of the wavelengths $\lambda_{\text{meas}}$ of the detected peaks obtained from the grating-based Ti:Sapphire scan as presented in Figure 4. Assignments from the literature are also given [34,35], along with level energies and configurations. The reported wavelengths are in vacuum.

| $\lambda_{\text{meas.}}$ (nm) | $\lambda_{\text{lit.}}$ (nm) | $E_i$ (cm$^{-1}$) | Configuration | $J_i$ | $E_f$ (cm$^{-1}$) | Configuration | $J_f$ |
|---|---|---|---|---|---|---|---|
| 420.72 | 420.712 | 6313.866 | $5f^6 6d7s^2\ ^7K_4^o$ | 4 | 30,083.102 | - | 5 |
| 420.77 | 420.767 | 0 | $5f^6 7s^2\ ^7F_0$ | 0 | 23,766.139 | $5f^6 7s7p\ ^7D_1^o$ | 1 |
| 420.94 | 420.942 | 2203.606 | $5f^6 7s^2\ ^7F_1$ | 1 | 25,959.849 | $5f^6 6d^2 7s$ | 1 |
| 422.32 | 422.306 | 4299.659 | $5f^6 7s^2\ ^7F_2$ | 2 | 27,979.161 | $5f^6 7s7p$ | 2 |
| 422.53 | 422.528 | 2203.606 | $5f^6 7s^2\ ^7F_1$ | 1 | 25,870.685 | $5f^6 6d^2 7s$ | 2 |
| | 422.539 | 6144.515 | $5f^6 7s^2\ ^7F_3$ | 3 | 29,810.974 | $5f^6 7s7p$ | 3 |

The presence of both lasers not only results in a higher background count rate, but also considerably enhances the count rates of the three most intense peaks. Similar to the hypothesis made with regard to the observation of a considerably higher ionization rate seen in the second experiment (a potential enhancement of the population of the 2203.61 cm$^{-1}$ state from the $^7D_1^o$ level), we suggest that these additional low-lying states are also populated via collisional de-excitation from higher-lying states, initially accessed via the ground-state transition.

In addition to the transitions mentioned in Table 1, a careful literature search revealed a candidate for excitation from the $5f^67s^2\ ^7F_6$ level at 10,238.473 cm$^{-1}$ to the $5f^67s7p$ J = 5 level at 34,004.30 cm$^{-1}$. This energy difference would result in a resonance at a wavelength of 420.772 nm, which lies only 8 GHz from the the original 420.767 nm ground-state transition. Due to the convolution of the frequency-doubled laser linewidth (>5 GHz) with the Doppler broadening of atomic lines in the gas cell, we expect the atomic resonances to have a broadening of >8 GHz. Interestingly, if the $^7F_6$ level is populated through collisional de-excitations from the $^7D_1^o$ level, the non-resonant ionization of plutonium can then proceed from the 34,004.30 cm$^{-1}$ level. This provides an alternative explanation to the ionization with a solely 420.77 nm laser light via an unknown Rydberg level. Due to the spectroscopic linewidth of the atomic transitions in the gas cell, we have a strong preference for this explanation.

Combining all of the additional spectroscopic information gathered with the grating-based Ti:sapphire laser, an extended ionization scheme for plutonium is presented in Figure 5.

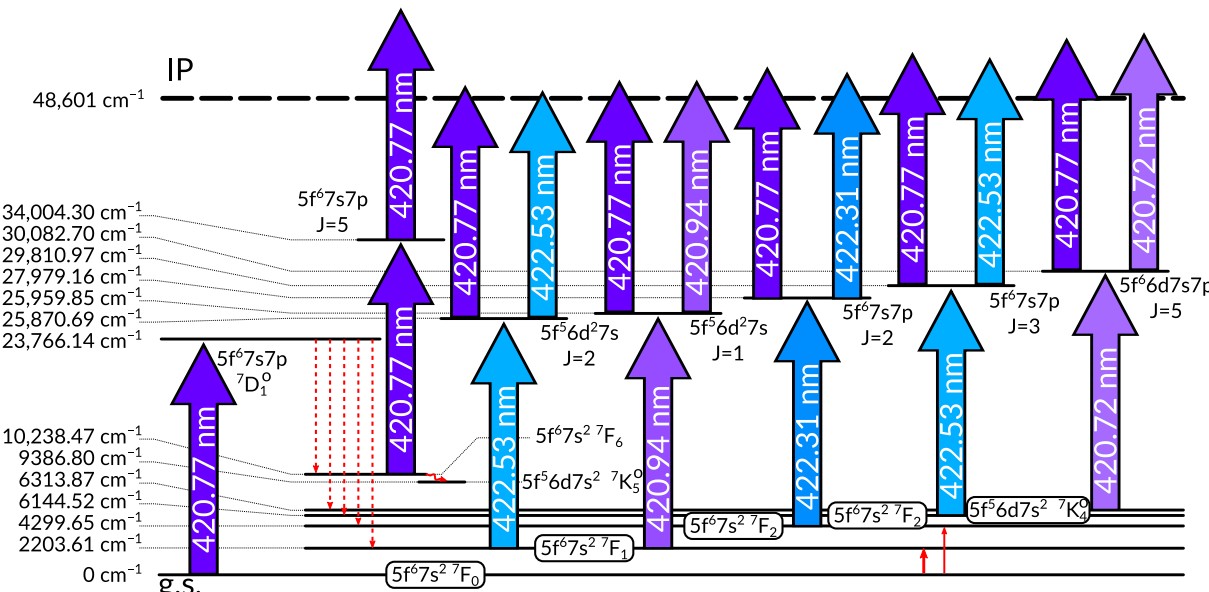

**Figure 5.** Plutonium atomic-level scheme developed with the grating-based Ti:Sapphire experiment. Thick arrows correspond to detected transitions, with the dashed arrows from the $^7D_1^o$ state representing the collisional de-excitation process to the low-lying states. The thin solid arrows starting from the atomic ground state indicate thermal excitation due to the hot filament.

## 4. Outlook

This work has shown evidence of collision-induced phenomena in plutonium through filament-based resonance ionization studies. A clear enhancement in the ionization rate of mass-separated $^{244}$Pu is observed when the ground-state transition to the excited $^7D_1^o$ level is added to a laser resonance ionization scheme that relies on a single-color photon exciting (and subsequently ionizing) from thermally populated low-lying levels. The lifetime of the $^7D_1^o$ state in the gas-cell environment appears to be much shorter than the rate for the photoabsorption of an IR second step of the original three-step scheme that was dominant in a vacuum environment. There is strong evidence therefore that the $^7D_1^o$ state is de-excited by inelastic collisions.

The exact nature of this phenomena is difficult to fully characterize, in particular in actinide elements in which the electronic level density is high. Very similar effects to that discussed here were observed in our work on thorium, in which laser resonance ionization was performed in both vacuum and in a helium buffer gas environment [23]. Similar to plutonium, in a reported work of the laser ionization of actinium, a reduced in-gas-cell

laser ionization efficiency was observed when compared with in-gas-jet ionization, which could partly be attributed to collisional quenching [36]. No experimental collisional cross section data for the quenching of excited levels for the actinides have been reported, to our knowledge. Nevertheless, in comparing the ionization schemes presented in this work, it is probable that the rate of collisional de-excitation is comparable to or greater than optical absorption rates.

In the near future, quantitative measurements will continue these investigations in neighboring uranium, both to be performed in vacuum and in the gas cell. Important information regarding the lifetimes of the excited state in the different environments will add to our understanding of the collisional processes. We note that despite the advantages of the gas-cell technique compared with the traditional ISOL method in terms of chemical non-selectivity, the buffer gas environment presents effects that need to be understood. Laser resonance ion sources using hot cavities have reported high efficiencies of ionization schemes for various elements. In the adaptation of these schemes for in-gas-cell resonance ionization, consideration should be given to the potential for very fast collisional channels that may well be detrimental to the efficiency of the method.

**Author Contributions:** Formal analysis, I.P.; Investigation, I.P.; Supervision, I.D.M.; Writing—original draft, A.R.; Writing—review & editing, A.R., I.P. and I.D.M. All authors have read and agreed to the published version of the manuscript.

**Funding:** This project has received funding from the European Union's Horizon 2020 research and innovation programme under grant agreement no. 861198–LISA–H2020-MSCA-ITN-2019, as well as from the Academy of Finland under project number 339245.

**Institutional Review Board Statement:** Not applicable.

**Informed Consent Statement:** Not applicable.

**Data Availability Statement:** The data presented in this study are available upon request from the corresponding author.

**Conflicts of Interest:** The authors declare no conflict of interest.

## Abbreviations

The following abbreviations are used in this manuscript:

| | |
|---|---|
| RIB | Radioactive Ion Beam |
| ISOL | Isotope Separator On-Line |
| IGISOL | Ion Guide Isotope Separator On-Line |
| LISOL | Leuven Isotope Separator On-Line |
| SHG | Second Harmonic Generation |
| SPIG | Sextupole Ion Guide |
| MCP | Multichannel Plate |
| FC | Faraday Cup |
| RIS | Resonance Ionization Scheme |

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
