# Peer review of "Observation of Collisional De-Excitation Phenomena in Plutonium"

_atoms, doi:10.3390/atoms10020040_

Round 1

Reviewer 1 Report

The manuscript "Observation of collisional de-excitation phenomena in actinide elements" by Andrea Raggio et al. investigate the potential reduction of the laser ionization efficiency through collisional de-excitation. This effect, which may be enhanced in elements with high atomic level densities such as the lanthanides or actinides, is investigated here via laser resonance ionization studies, on the case of plutonium. The article is well written, useful and appropriate for publication in "Atoms". However, the title is not adequate, since in the article, the investigation 
has been performed on plutonium. So plutonium must be present in the title. One way is for example: "... elements: the case (or the example or something similar) of plutonium", or "... phenomena in plutonium, as a representative of actinides", or something similar. 

Reviewer 2 Report

Manuscript report

Title: “Observation of collisional de-excitation phenomena in actinide elements”

Authors: Andrea Raggio, Ilkka Pohjalainen and Iain D. Moore

Manuscript number: atoms-1677084

In the submitted paper the Authors report results on laser ionization of plutonium atoms. Their research activity fits in the framework of high-resolution optical spectroscopy of the actinide elements made at IGISOL facility of the University of Jyvaskyla in Finland. In particular the Authors focus on the enhancement effect on ionization rate arising from the collision de-excitation induced phenomena.

In the manuscript, the authors use a three steps ionization method by laser radiation in order to populate high level of plutonium atoms and surprisingly they found that two of the three steps do not contribute to the ionization rate as a result of the wavelength detected experimentally. The authors attribute this behaviour to the de excitation processes among the atomic levels due to the collisions with He atoms.

Theoretically speaking, and for a future development, a similar behaviour could be quantitatively understood in a state-to-state kinetic approach for the energy levels of plutonium in a similar way reported for example in the following papers concerning vibrational levels excitation for molecules:

V. Laporta, K.L. Heritier and M. Panesi, “Electron-vibration relaxation in oxygen plasmas”, Chemical Physics 472, 44-49 (2016)

K.L. Heritier, R.L. Jaffe, V. Laporta and M. Panesi, “Energy transfer models in nitrogen plasmas: Analysis of N2(X S1+g) N(4Su) e interaction”, J. Chem. Phys. 141, 184302 (2014)

This approach should be mentioned in the paper.

The manuscript, in general, is well presented, the experimental set up analytically described and the results correctly analysed. I think that the paper itself is up to the standard of the journal ‘atoms’ and it is well aligned with the scope of the journal. Finally, I believe that the results are very interesting for the spectroscopy community.

For all these reasons I strongly suggest for the publication of the paper.
